# The Effect of Sodium Total Substitution on the Quality Characteristics of Green Pickled Tomatoes (*Solanum lycopersicum* L.)

**DOI:** 10.3390/molecules27051609

**Published:** 2022-02-28

**Authors:** Gabriel-Dănuț Mocanu, Oana-Viorela Nistor, Oana Emilia Constantin, Doina Georgeta Andronoiu, Viorica Vasilica Barbu, Elisabeta Botez

**Affiliations:** Department of Food Science, Food Engineering, Biotechnology and Aquaculture, Faculty of Food Science and Engineering, “Dunărea de Jos” University of Galați, 800201 Galați, Romania; danut.mocanu@ugal.ro (G.-D.M.); emilia.constantin@ugal.ro (O.E.C.); georgeta.andronoiu@ugal.ro (D.G.A.); vasilica.barbu@ugal.ro (V.V.B.); elisabeta.botez@ugal.ro (E.B.)

**Keywords:** pickling, antioxidant activity, microbiota, texture, sensorial

## Abstract

Green pickled tomatoes are a traditional fermented product in Romania. This study was focused on the effect of total substitution of NaCl with KCl and MgCl_2_ on physicochemical and microbiological quality; bioactive compounds; and microstructural, textural, and sensorial properties of fresh and pickled green tomatoes during 28 days of fermentation. By the means of physicochemical composition, the NaCl addition induced the most stable characteristics for the pickles compared to the other two types of salts. The content of total flavonoids in green pickled tomatoes with NaCl (34.72 ± 0.43 mg CE/g DW) was significantly lower compared with the control sample (63.80 ± 0.55 mg CE/g DW). The total number of lactic acid bacteria (LAB) at the final stage of fermentation varied between 4.11 and 4.63 log CFU for all variants. The textural analysis revealed that the NaCl has the lowest influence on the textural parameters. Finally, the overall acceptance of green pickled tomatoes containing KCl and MgCl_2_ was found to be proper to be consumed as a substitute for pickles with NaCl.

## 1. Introduction

Fruits and vegetables are the most nutritious and perishable foods that have a very low shelf life. Different methods of preservation have been practiced for centuries. The oldest preservation method of several foodstuffs, such as fruits, vegetables, meats, and fish, is pickling in brine and/or vinegar. This preservation method was an intrinsic procedure in all communities and cultures over the entire world [1]. There are a various types of pickles that are divided into two categories, depending on the fermentation process [2]:
-Unfermented pickles consist of (i) “salt-stock pickle” where preservation is due to salt and not to fermentation process; and (ii) “vinegared pickles”, using acids (wine and/or vinegar (acetic acid)), salt, and sometimes sugar to obtain a pleasant flavor and texture in pickles.-Fermented pickles, often called brine pickles, are divided into sour and sweet pickles: (i) sour fermented pickles who belong to the products stabilized with a dilute brine (2–5% salt); and (ii) sweet fermented pickles obtained by using a combination of acetic or lactic acid, sugar and spices, or aromatic herbs [1].

The tomato (*Solanum lycopersicum* L.) belongs to the Solanaceae family and is a popular and versatile fruit, being consumed by people all over the world, fresh and processed as sauce, paste, pickled, juice, dried, or concentrate [3,4]. Tomatoes are considered to be a nourishing food that is rich in vitamins, minerals, proteins, dietary fibers, and a multitude number of bioactive and valuable compounds, such as carotenoids (lycopene, phytoene, and *β*-carotene) and polyphenols (flavonoids, flavanones, and flavones) that provide health benefits [5]. Numerous studies have demonstrated that the consumption of tomatoes has health properties against many types of cancer [6], slowing the cardiovascular disease and hypertension [7], anti-inflammatory [8], anti-genotoxic, and anti-mutagenic effects [9] due to the bioactive compounds. In contrast with the typical red tomatoes, green (unripe) tomatoes contain high levels of glycoalkaloids, such as tomatine, tomatidine, and tomatidenol. Tomatidine has many beneficial properties, which include cardioprotective, anti-inflammatory and antioxidative effects [10,11].

Green pickled tomatoes are a traditional fermented product in Romanian cuisine used especially during the cold period of the year. The fermentation process of this type of product is easy because we do not need distinctive equipment. The indispensable or essential ingredients to obtain this Romanian traditional product are salt, sugar, vinegar, water, and spices or aromatic herbs. Green pickled tomatoes can include acetic acid or lactic acid, which acts as preservatives to maintain a safe and quality product for a long period.

Despite the multiple health benefits, pickles are sometimes indictable for the high content of salt. Closely related to the high dietary salt intake is the occurrence of hypertension, which is the leading cause of premature death worldwide. Therefore, the World Health Organization has recommended that sodium intake should not exceed 2 g per day, which corresponds to about 5 g of salt intake [12]. The total or partial substitution of sodium chloride (NaCl) with potassium chloride (KCl), calcium chloride (CaCl_2_), magnesium chloride (MgCl_2_), or zinc chloride (ZnCl_2_) was successfully used to vegetable fermentation. Several studies have been carried out to examine this effect on cucumber and carrot pickles produced with KCl, CaCl_2_, or MgCl_2_ [13,14]; mango pickles obtained with calcium and potassium salts [15]; and low-sodium lime pickles [16]. In these studies, the chemical composition, microbiological, and sensorial analysis, as well as texture profile, were investigated.

The main objective of the present study was to evaluate the effect of total substitution of sodium chloride with potassium chloride and magnesium chloride in the case of green pickled tomatoes. These fermented pickles were analyzed for carotenoids content (lycopene and *β*-carotene), polyphenols, flavonoids, antioxidant activity, lactic acid, ash and salt content, and microbiological and sensory qualities. The structural and morphological particularities of green pickled tomatoes were analyzed by using confocal laser microscopy. Textural properties were also analyzed to characterize this Romanian traditional product.

## 2. Materials and Methods

### 2.1. Materials and Chemicals

Green tomato (*Solanum lycopersicum* L.) fruits were obtained from a local market in Galați, Romania and stored at 4 °C before the analysis. The drying weight content was determined with XM Precisa moisture analyzer (Switzerland) to the value of 4.95 ± 0.3%. The salts, sodium chloride (Salrom, Târgu Ocna, Romania) potassium chloride (S.C. Amniocen S.R.L., Timișoara, Romania), and magnesium chloride (BioSano, Piatra Neamț, Romania), were provided from a local market in Galați, Romania.

The chemicals used for chemical analyses, namely 2,2′-Azino-bis(3-ethylbenzothiazoline-6-sulfonic acid) diammonium salt (ABTS), 6-hydroxy-2,5,7,8-tetramethylchromane-2-carboxylic acid (Trolox), ethanol, ethyl acetate, methanol, beta-carotene, lycopene, Folin-Ciocalteu’s reagent, sodium carbonate, aluminum chloride, gallic acid, quercetin, lactic acid, iron (III) chloride, Plate Count Agar, Rose Bengal Chloramphenicol Agar medium, MRS agar, MRS broth, were purchased from Sigma-Aldrich (MilliporeSigma, Steinheim, Germany).

### 2.2. Procedure for Green Tomato Pickles Preparation

Fresh green tomatoes were washed with tap water, cut into small pieces, and placed in fifteen sterilized jars (five sterilized jars for each type of green pickled tomatoes). The marinade was prepared by boiling about 1.5 L of tap water with 150 g vinegar (acetic acid, 9°), 37.5 g sunflower oil, 112.5 g sugar, 105 g celery leaves, 60 g onions, 6 g garlic, 0.3 g bay leaves, 0.6 g allspice corns, and 0.6 g whole black peppercorns. The obtained marinade was divided into three equal parts, and the salts (sodium, potassium, and magnesium chloride—9 g from each type) were dissolved. Three variants of green pickled tomatoes were obtained, coded as follows: P_Na_—green pickled tomatoes with sodium chloride, P_K_—green pickled tomatoes with potassium chloride, and P_Mg_—green pickled tomatoes with magnesium chloride. The jars were tightly closed and stored at ambient air temperature, 16–18 ± 2 °C, for 28 days. The first samples were taken at seven days of pickling. Other samples were taken on the 14th, 21th, and 28th day of fermentation. The fresh green tomatoes were considered the control sample.

### 2.3. Chemical Analyses

#### 2.3.1. Moisture and Protein Content

The moisture and protein contents of the green pickled tomatoes were determined by using the standard method [17].

A sample of 10 g was put in a clean Petri dish and exposed to a hot-air oven at 70 °C till reaching the constant weight. The loss of weight was expressed as moisture fraction of initial weight, according to Equation (1):(1)$$M_{f}=\frac{W_{bd}-W_{ad}}{W_{bd}},\;\%$$
where M_f_ is the moisture fraction, %; *W_bd_* is the sample weight before drying, g; and *W_ad_* is the sample weight after drying, g.

The protein content was determined by the Kjeldahl method [18] from total nitrogen, using a factor of 6.25. The calculation formula is presented in Equation (2):Crude protein = Crude nitrogen × 6.25, %(2)

#### 2.3.2. Salt Content

The salt content of the brine was determined directly with a B-722 Sodium Ion Analyzer (Horiba Instruments, Singapore).

#### 2.3.3. Ash Content

A total of 1 g of sample was weighted to be used in the analysis of ash content. The pickles were finely ground and carefully mixed. The samples were heated for 30 min in a muffle furnace set at 550 ± 25 °C. The samples were then transferred into desiccators and cooled for 30 min; they were then weighed until the two recent weight difference were within 0.5 mg. According to Reference [19], the ash content can be expressed by the following calculation:(3)$$\%\;\mathrm{Ash}=\frac{M_{Ash}}{M_{wet}}\cdot \text{100}$$
where *M_Ash_* refers to the mass of the ashed sample, and *M_wet_* refer to the original masses of the wet samples.

#### 2.3.4. Lactic Acid Determination

The lactic acid in brine was determined by separating the liquid from the cells by centrifuging. The supernatant was diluted 20-fold with deionized water. A spectrophotometric method for the determination of lactic acid was used. The concentration of lactic acid was calculated by using a calibration curve, taking into account the 20-fold dilution of the test sample [20]. The calibration curve was determined by using a series of stock lactic acid solutions prepared by two-fold dilutions. Therefore, 1.2 g of lactic acid with the known concentration of 89%, ρ = 1.2 g/mL, was placed in a 10 mL volumetric flask and diluted with water. A stock solution with the x concentration of lactic acid 89 g/L was obtained. The sequel of the experiment is described for the samples determination.

For the spectrophotometric determination of lactic acid, a test solution (50 μL) containing lactic acid was added to 2 mL of a 0.2% solution of iron (III) chloride and stirred, and absorbance was measured at 390 nm against the reference solution (2 mL of a 0.2% FeCl_3_ solution). The reaction and measurements were performed at 25 ± 5 °C. The color of the solution was stable for 15 min. The experiments were performed in triplicate.

### 2.4. Enumeration of Mesophilic Aerobic Bacteria (MAB), Lactic Acid Bacteria (LAB), and Yeast (Y) Values

#### 2.4.1. Mesophilic Aerobic Bacteria Count

The method involved inoculating the corresponding dilutions on the Plate Count Agar (PCA) medium, and then the plates were incubated at 30 ± 1 °C for 72 h. After incubation, the colonies were counted, and the colony-forming units were determined per gram product and expressed as log CFU/mL brine [21].

#### 2.4.2. Yeasts Count

The determination was performed by cultivation on Rose Bengal Chloramphenicol Agar medium, incubated at 25 °C, for 3–5 days [22]. The results were expressed as log CFU/mL brine.

#### 2.4.3. Lactic Acid Bacteria Count

The LAB were determined by cultural method, on double-layer MRS agar [23], in duplicates, using appropriate dilution, over 28 days. Briefly, different dilutions were achieved, and from each, a volume of 100 μL was spread on MRS supplemented with 1.5% agar and 1% CaCO_3_. A second layer of the medium was added (5 mL MRS with 0.75% agar and 1% CaCO_3_), and the plates were incubated at 35 ± 2 °C for 72 h. The results were expressed as log CFU/mL brine.

### 2.5. Ultrasound-Assisted Extraction (UAE) of Carotenoids

UAE was performed by using an ultrasonic bath system (MRC Scientific Instruments, Harlow, UK). A total of 1 g of chopped pickles was mixed with 10 mL of solvent (ethyl acetate) and then introduced into an ultrasonic bath equipped with a digital control system of sonication time, temperature, and frequency. The extraction process was performed at a constant frequency of 40 kHz, with a constant power of 100 W for 30 min. To maintain a constant temperature (t = 40 ± 3 °C) in the ultrasonic bath, cold water was added. After the extraction, the supernatant was separated by centrifugation at 9000 rpm for 10 min.

### 2.6. Determination of Beta-Carotene and Lycopene Content

The obtained extracts were diluted in the solvent used for extraction (ethyl acetate) in order to measure the absorbance at different values of the wavelength: λ = 450 nm (*β*-carotene) and λ = 503 nm (lycopene). The content of carotenoids (CC) was calculated according to the following equation [24]:(4)$$\mathrm{Carotenoid}\;\mathrm{Content}\;(\mathrm{mg}/g)=\frac{A\cdot V\cdot 10^{6}}{\varepsilon \cdot m\cdot 100}\cdot f_{d}$$
where *A* is absorbance; *V* is the final volume of the analyzed extract, in mL; *ε* is the extinction coefficient (*ε* = 2500 for *β*-carotene; *ε* = 3450 for lycopene); *m* is the weight of the sample, in grams; and *f_d_* is the dilution factor. Carotenoid content is expressed in milligrams per gram sample DW (mg/g DW). The experiments were performed in triplicate.

### 2.7. Determination of Total Polyphenol and Total Flavonoid Content

To determine the calibration curve for total polyphenol content, 1 mL of standard solution of gallic acid (20, 40, 60, 80, and 100 mg/L) was added to 9 mL of distilled deionized water. A total of 1 mL of Folin–Ciocâlteu reagent was added to the mixture and shaken for 5 min. Then 10 mL of 7%Na_2_CO_3_ solution was added to the mixture. After that, distilled deionized water was added until the volume of 25 mL and mixed. After 90 min in the dark at room temperature, the absorbance was determined at 765 nm [25].

The total polyphenol content (TPC) was measured by using the Folin–Ciocâlteu colorimetric method. Then 0.5 mL of the obtained extracts was introduced into test tubes and mixed with 2.5 mL of 1:10 diluted Folin–Ciocâlteu reagent and 2 mL of 7.5% sodium carbonate. The test tubes were covered with aluminum foil and then incubated at room temperature for 30 min. The absorbance was measured at 765 nm with an UV–Vis spectrophotometer (Biochrom Libra S22 UV/Vis, Cambridge, UK) [26]. The results were expressed as mg of gallic acid equivalents (GEA) per gram of sample DW (mg GAE/g DW). All determinations were performed in triplicate.

The calibration curve was determined by using an aliquot of 1 mL of standard solution of catechin (20, 40, 60, 80, and 100 mg/L), to which 4 mL of distilled deionized water was added. Then 0.3 mL NaNO_2_ of 5% concentration was added, and after 5 min, 0.3 mL 10% AlCl_3_ was added. After 6 min, the solution was mixed with 2 mL 1 M NaOH, and the total volume was brought to 10 mL with distilled deionized water. The absorbance was determined at 510 nm [25].

The total flavonoid content (TFC) was determined by using the aluminum chloride colorimetric method, according to Reference [25]. To 0.5 mL of the obtained extracts, 0.6 mL of 2% aluminum chloride was added. After 60 min of incubation at room temperature, the absorbance of the mixture was measured at 420 nm with an UV–Vis spectrophotometer (Biochrom Libra S22 UV/Vis, Cambridge, UK). The results were expressed as mg of catechin equivalents (CE) per gram of sample DW (mg CE/g DW). All determinations were performed in triplicate.

### 2.8. Antioxidant Activity Assay

The TEAC assay of the ultrasound-assisted extracts dissolved in ethyl acetate was performed according to the method described by Reference [27]. The results were expressed as mmol Trolox equivalents (TE) per gram of sample DW (mmol TE/g DW). The experiments were performed in triplicate.

### 2.9. Structural and Morphological Properties of Pickled Green Tomatoes

In this study, we compared, by using confocal laser scanning microscopy (CLSM), the microstructure of the plant tissues from tomato fruit (*Solanum lycopersicum* L.) preserved by three different methods, using NaCl (P_Na_), KCl (P_K_), and MgCl_2_ (P_Mg_) in order to capture the textural changes. The confocal microscopes Zeiss LSM 710 (Carl Zeiss MicroImagining, Göttingen, Germany) work on the principle of point excitation in the specimen and point detection of the emitted fluorescent signal. For excitation, the confocal laser system is equipped with the following lasers: diode laser (405 nm), Ar-laser (458, 488, and 514 nm), DPSS laser (diode pumped solid state (561 nm) and HeNe-laser (633 nm). The images were captured with an AxioObserver Z1 inverted microscope, endowed with 20× fluar objective (numerical aperture 0.4) and the FS49, FS38, and FS15 filters. ZEN 2012 SP1 software (black edition) (Carl Zeiss MicroImagining, Göttingen, Germany) was used for image analysis. The acquisition parameters of the 3D images were line scan mode, mean method, speed 6, 12-bit depth, and a frame average of eight scans, in order to increase the signal-to-noise ratio. The samples were examined in their native state or by staining with two dyes, namely DAPI (1 μg/mL) and Red Congo (40 μM/L), in a ratio 1:1.

### 2.10. Textural Analysis

Texture analysis was achieved with a Brookfield CT3 Texture Analyser (Brookfield Ametek, Middleboro, MA, USA). The green tomatoes (raw and after 7, 14, 21, and 28 days of pickling) were subjected to a double compression test, using a 50.8 mm–diameter probe. The test speed was set at 1 mm/s, target distance was 1.5 mm, trigger load was 0.067 N, and load cell was 1000 g. The data were registered and processed with TexturePro CT V1.5 software, revealing four textural parameters: firmness, cohesiveness, springiness, and chewiness.

### 2.11. Sensory Analysis

The sensorial analysis of the green pickled tomato samples was performed by using an Acceptance Test, which refers to the following sensorial attributes: appearance, saltiness, sweetness, sourness, spiciness, mouthfeel, texture (hardness and cohesiveness), color, and overall acceptability. The Acceptance Testing was performed by using a hedonic scale of 9 points (1 represents dislike extremely, while 9 represents like extremely). Before sensorial analysis, the product samples were left at room temperature (18…20 ± 2 °C) for 1 h to bring the pickles to the consumption temperature, then sliced to approximately equal sizes, placed on white plates, and coded with random numbers. Pickles’ evaluation was performed by 15 untrained panelists (5 males and 10 females) in the age range 20–22 years old. The main criterion for the panelists’ choice was “regular consumption of pickles”. Sensory evaluation of each fermented sample was carried out after 28 days.

### 2.12. Data Analysis

The measurements were performed in triplicate for each sample and the results were reported as mean ± standard deviation (SD). Data were analyzed by using Minitab for windows version 19.0. One-way analysis of variance (ANOVA) was used to identify significant differences between experimental data obtained for pickle samples. The Tukey test with a 95% confidence interval was applied when significant differences were observed; *p* < 0.05 was considered to be statistically significant.

## 3. Results and Discussion

### 3.1. Physicochemical Properties of Fresh and Pickled Green Tomatoes

Moisture and protein contents, as well as ash, were determined for the control sample, represented by the raw tomato (Table 1). Three varieties of green pickled tomatoes were analyzed in terms of moisture, ash, salt, protein, and lactic acid content. All the determinations were registered for 0 (control sample), 7, 14, 21, and 28 days of storage, in the same conditions, at 16–18 °C.

A slightly increase of the moisture content was registered by the sample with NaCl, while, for the samples preserved with KCl and MgCl_2_, no significant changes were determined. The ash content ranged from 0.61 ± 0.04% to 0.98 ± 0.07% for all the samples, except for the control sample, with 0.37 ± 0.02%. The ash values were stable for the samples preserved by using NaCl, while the most important change was registered by the KCl samples, followed by the MgCl_2_ samples. These may be a consequence of the diffusion phenomenon when the cellular tissues are soaked and the constituents could migrate into brine.

While the initial percentage of added salt was of 1%, the pickles registered values between 0.55 ± 0.04% and 0.74 ± 0.06%, but this is not such a significant variation. Moreover, it seems that regardless of the type of salt there were not significant variations in the green pickled tomatoes. Moreover, it could be seen that the NaCl generates a different kind of absorption compared to the other samples. After 14 days of fermentation the salt content percentage is constant until 28 days, so the content varied between 0.55 ± 0.04% for 7 days of storage and 0.74 ± 0.06% for 28 days of fermentation. For the samples preserved by using KCl or MgCl_2_, the situation is not similar, and the salt content is decreasing during the storage from 0.7 ± 0.05% to 0.57 ± 0.06% for KCl samples, while for MgCl_2_ samples, the salt content (0.66 ± 0.02% and 0.62 ± 0.04%) is almost the same. Similar results were obtained by Reference [28] for some vegetable pickles.

The protein content of raw green tomatoes ranged between 0.2 and 1.50 g/100 g in the present study (1.50 ± 0.06%). During the fermentation process, the protein content is slightly affected by the salts addition and the fermentation process. For the NaCl addition, wenote a difference in behavior of the samples compared to the other two types of salt, whereby the increase in protein value is constant after the first 7 days of fermentation. A similar evolution of the data was reported by Reference [29] in a study on fermented pea protein.

The natural fermentation of vegetables is due primarily to the activity of naturally occurring lactic acid bacteria and other diverse microorganisms, such as yeasts and filamentous fungi [30]. These are involved in several stages of fermentation and could produce lactic acid and other important metabolites surviving to high levels of lactic acid conditions.

The producing of lactic acid in pickles could be characterized by four stages, namely initiation, primary fermentation, secondary fermentation, and post-fermentation spoilage, as [31] were reported in their study on fermented sauerkraut. Correlated with the other results of the study, it seems that the green pickles’ fermentation period is 28 days, but the production of lactic acid content strongly increased until the 14th day of fermentation.

The lactic acid production is dependent on the salt-type addition and the substrate composition. The NaCl induces a lactic acid increase of almost 200%, while the MgCl_2_ a 300% increase in the highest fermentation phase up to 14 days of fermentation. After that period, a significant decrease in lactic acid content between 64 and 70% in the 21st day of fermentation can be observed. Similar results were obtained by Reference [32] for pickled radish. Even while the lactic acid production is decreasing, other important metabolites are achieved by the green pickled tomatoes, and this affirmation is sustained by the phytochemical results.

### 3.2. Effects of Pickling on Beta-Carotene, Lycopene, Total Polyphenol and Total Flavonoid Content

In Table 2 are presented the main phytochemicals determined for the fresh and pickled tomatoes during the fermentation period of 28 days.

The bioactive compounds for fresh green tomatoes, as well as for the three types of salted pickles, were determined and reported as *β*-carotene and lycopene content, total phenolic content (TPC), and total flavonoid content (TFC).

It could be seen that the *β*-carotene and lycopene contents are strongly correlated, and the values are increasing during the fermentation period of 28 days. The values for *β*-carotene content of the green pickled tomatoes fermented with NaCl are six times higher than the control sample value, while, for those with KCl and MgCl_2_, the values are 5, respectively 4 times higher. It seems that the NaCl addition had a positive effect on the *β*-carotene retention in green pickled tomatoes during the fermentation period. These findings are similar to Reference [33] for 10 vegetables and Reference [34] for orange fleshed sweet potato, which have reported the improvement of the levels of bioactive compounds and antioxidant capacities of pickled vegetables after 30 days of fermentation. Depending on the type of salt, this could help dissociate the carotenoid protein complexes into the brine or maintain them in the pickles.

Lycopene is the major component of carotenoids and the increasing of the content in green pickled tomatoes is highly correlated to the *β*-carotene content. The lycopene content increases constantly during fermentation. The highest values (P_Na_-162.82 ± 0.82 mg/g DW, P_K_-172.67 ± 0.23 mg/g DW, and P_Mg_-124.76 ± 0.80 mg/g DW) were determined for 28 days of fermentation. Consequently, the samples with a NaCl and KCl addition have a 5-fold increase in lycopene content, while the samples with MgCl_2_ had a 4-fold increase.

After the first stage of fermentation of 7 days, the values for the TPC and TFC of the pickles with NaCl were decreased by almost 35%, with respectively 85% reported to the control samples. Significant decreases are also reported for the pickles with potassium and magnesium chloride. This phenomenon could be partially attributed to the migration of the phenols into the brine and partially to the different biochemical, physiological, and structural reactions that occur during the first stage of the fermentation process. Similar results were obtained by Reference [35] for the fermented red radish.

After 14 days of fermentation and up to the final stage of 28 days, a significant increase in TPC (59.42 ± 0.71–128.03 ± 0.49 mg GAE/g DW for the pickles with NaCl, 57.65 ± 0.85–72.57 ± 0.91 mg GAE/g DW for the pickles with KCl, and for the pickles with MgCl_2_ 57.88 ± 1.09–72.79 ± 0.92 mg GAE/g DW) and TFC (20.23 ± 0.56–34.72 ± 0.43 mg CE/g DW for the pickles with NaCl, 2.81 ± 0.14–11.87 ± 0.52 mg CE/g DW for the pickles with KCl and for the pickles with MgCl_2_ 1.81 ± 0.14–6.42 ± 0.79 mg CE/g DW) was registered for all the samples.

In the presented experiment, it was found that the concentration of total flavonoids in green pickled tomatoes with NaCl (34.72 ± 0.43 mg CE/g DW) was significantly lower compared with the control sample (63.80 ± 0.55 mg CE/g DW). A reduction of almost 82% and 90% was registered for the samples fermented with KCl and MgCl_2_, respectively. Different studies [36,37] have addressed the divergency in the negative correlation between the TPC and TFC content. The agricultural factors remain the main variable, as well as the nitrogen content and availability, which is highly correlated with flavonoid concentration in fruits and vegetables. The fermentation conditions, such as temperature, light exposure, and type of salt, can cause variation in flavonoid concentration in green pickled tomatoes. Moreover, the structure, activity, and bioavailability of these compounds might change over-processing, as was pointed out by Reference [38], for the degradation of glycoside from onion bulbs.

### 3.3. ABTS-Radical Cation Scavenging Activity

The antioxidant activity of fresh and green pickled tomatoes was assessed by means ABTS method, and the results are presented in Figure 1. The ABTS^·^ radical scavenging activity of the fresh green tomatoes (control sample) was 23.95 ± 0.5 mmol TE/g DW. After the first 7 days of pickling, the ABTS^·^ radical scavenging activity ranged between 28.11 ± 0.75 and 37.05 ± 0.45 mmol TE/g DW.

The pickling process caused an increase of antioxidant activity between 14.8% and 35.36%, respectively, of green pickled tomatoes samples when compared to the control samples. The high antioxidant activity may be justified due probably to the increase in bioactive compounds content, such as carotenoids, polyphenols, and flavonoids. After 21 days of pickling, antioxidant activity continues to increase with a maximum of 36.32%, and this percent was established for the sample P_K-21_ (37.61 mmol TE/g DW) compared with the value of the fresh product.

Until the experimental period ends in 28 days, the results showed a significant increase of ABTS radical scavenging activity by 54.18–74.47%, respectively. We consider that the types of bioactive compounds that are dependent upon the raw materials, maturity, cultivars, and pickle processing method could influence the values of antioxidant activity.

### 3.4. Microbial Properties of Pickled Green Tomatoes

The initial population of fresh vegetables was approximately 1.31 log CFU/g. During pickling with NaCl, the mesophilic population varied from 1.91 and 3.66 log CFU/g throughout fermentation at room temperature. Similar results were obtained for the samples with KCl and MgCl_2_. The study aimed to identify changes in the microbiota through the pickling procedure during the fermentation period in terms of mesophilic bacteria, LAB, and yeast data shown in Table 3.

The yeast counts of the samples were found to be below the detection limit for 28 days of fermentation (<1 log CFU/g). Comparable results were obtained by Reference [39] for mixed pickles (Tursu).

Naturally occurring lactic acid bacteria from the green tomato pickles play an essential role in the fermentation process. The total number of LAB at the final stage of the fermentation period varied between 4.11 and 4.63 log CFU for all variants. The salt concentration determines lactic fermentation production by involving the lactic acid bacteria present on the vegetables used. Moreover, the adding salts are responsible for inhibiting the salt-sensitive spoilage bacteria.

### 3.5. Microscopic Observation of Green Pickled Tomatoes Tissue

Confocal laser microscopy analysis of green pickled tomato samples preserved by different methods highlighted the impact of salts on the tissue structure. In the presence of NaCl, the best texture was preserved (Figure 2) for P_Na_ native sample; large parenchymal cells (125.49–134.70 µm) can be observed, with intact cell walls (CW), rich cytoplasmic content, nuclei (N) in blue, numerous chloroplasts (Ch) in green, and inclusions—probably lycopene (I) in red.

Fragments of vascular tissue, xylem vessels (X) with spiral or scalar ornamentations (Figure 2), in the P_Na_-stained sample can be observed frequently. KCl affected most of the structure of the plant tissue; thus, there could be frequently observed deformations and lysed (L) cells, with smaller chloroplasts, aggregates, and acicular lycopene for the P_K_ sample both native and stained. The sections made in tomatoes pickled with MgCl_2_ (Figure 2 P_Mg_ native sample) showed fragments of parenchymatic tissue of normal constitution, whose cells have an average size of 16.87–17.83 μm. It is noteworthy that the presence of large spherosomes up to 112.88 μm (in red) probably resulted from the interaction of magnesium chloride with cell inclusions (Figure 2 P_Mg_ native and stained sample).

### 3.6. Texture Quality

The results of the textural analysis are presented in Figure 3. For firmness, expressed as the maximum compression force registered for the first cycle, a continuous decrease during pickling was observed. This behavior is due to the action of pectinolytic enzymes. The most accentuated texture softening was remarked for the samples containing magnesium chloride (from 7.26 N to 3.01 N), while the natrium chloride induced the lowest firmness loss (from 7.26 N to 3.86 N). This might be owned to the antimicrobial effect of NaCl, which inhibits the development of contamination microorganisms together with the production of exopolygalacturonase [40]. Similar values for firmness were reported by Reference [12] for low-sodium pickled cucumbers. Cohesiveness is a measure of the vegetal internal bonds strength [41]. For the pickled tomatoes samples, we noticed the reduction of cohesiveness from 0.67 for fresh samples to 0.5, 0.42, and 0.45 for 28 days of pickled samples in NaCl, MgCl_2_, and KCl brines, respectively.

The highest decrease of cohesiveness registered for the samples pickled in MgCl_2_ brine, which may show a breakage of the tissue cells. A similar evolution was noted for springiness, defined as the capacity of the sample to recover the deformation after the force is removed, and for chewiness, defined as the energy required for mastication before swallowing. The textural analysis revealed that the NaCl has the lowest influence on the textural parameters. In order to correlate the instrumental and the sensorial texture analysis, the linear correlation coefficients were determined. They registered values of 0.62 for firmness and 0.89 for cohesiveness.

### 3.7. Sensorial Characterization of Green Pickled Tomatoes

Figure 4 summarizes the sensory attributes of green pickled tomatoes after the fermentation for 28 days. The sensorial characteristics are placed on radial axes, with the intensity ranging from 0 (none) at the center of the circle to 9 (very strong) at the outer circumference. The mean scores for appearance ranged from 7.31 ± 1.16 (P_K_) to 7.38 (P_Na_ and P_Mg_). The quantity of sweetener (sugar) and acetic acid (vinegar) in green pickled tomatoes had an important effect on the sweetness and sourness of the samples. Sourness scores ranged from 5.56 ± 0.99 to 6.75 ± 1.16 and were higher than sweetness scores, which varied between 5.06 ± 1.52 and 6.12 ± 0.78. In this case, we can say that, probably, the quantity of sugar influenced the perception of sourness, and the vinegar affected the perception of sweetness. All pickled samples were found to be only very slightly salty. It is possible that this sensorial attribute was masked entirely by the quantity of added sweetener.

The spiciness was fluctuated between 4.18 ± 1.51 (P_K_) and 5.94 ± 1.03 (P_Na_). The mean scores for mouthfeel ranged from 6.13 ± 1.36 to 4.00 ± 0.94. The texture is a relevant characteristic for all pickles. For this aspect, hardness and cohesiveness differ between the samples, indicating that the total substitution of NaCl by KCl or MgCl_2_ affected these parameters. Thereby, the hardness and cohesiveness were higher for the samples pickled in a brine solution with NaCl or MgCl_2_ compared to the sample pickled with KCl.

The average values of overall acceptability were between 5.19 ± 1.01 and 6.81 ± 1.23 in the hedonic scale of 9 points. From these results, we can say that more than 65% of the panelists considered that pickles obtained with KCl or MgCl_2_ were acceptable for consumption.

## 4. Conclusions

The study confirmed that the substitution of NaCl in the production of green pickled tomatoes could be a technological alternative for traditional pickling, even though there are some limitations. It is worth pointing out that the use of different chloride salts as replacers for total substitution of NaCl with KCl or MgCl_2_ influenced the physicochemical and microbiological quality of green pickled tomatoes. However, the pickling process where KCl and MgCl_2_ are used for NaCl substitution is a proper method for the preservation with beneficial effects on the concentrations of bioactive compounds, such as carotenoids (*β*-carotene and lycopene), TPC, TFC, and on antioxidant activity after 28 days of fermentation. The results of sensory analysis prove that green pickled tomatoes fermented with KCl or MgCl_2_ are adequate for consumers who want to reduce the intake of sodium salts and to cure hypertension. The textural analysis revealed that the NaCl has the lowest influence on the textural parameters. In conclusion, all types of green pickled tomatoes can be considered acceptable and safe for consumption. Further studies are needed to better understand the role of K and Mg ions on these types of fermented products and their impacts on the pickles’ quality.

## Figures and Tables

**Figure 1 molecules-27-01609-f001:**
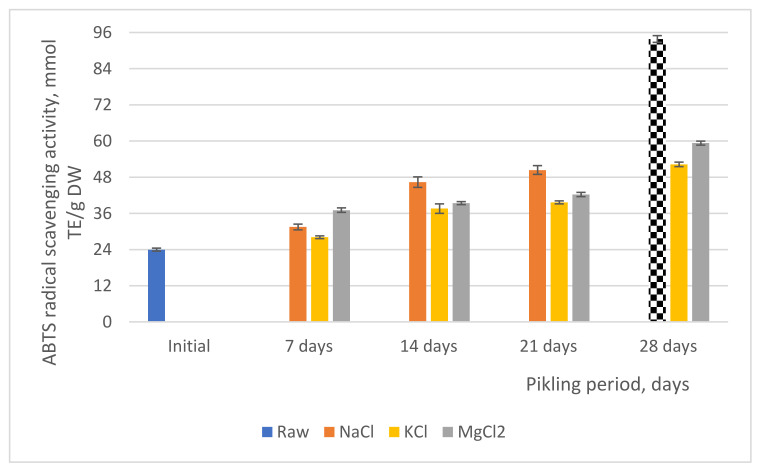
Antioxidant activity of fresh and green pickled tomatoes.

**Figure 2 molecules-27-01609-f002:**
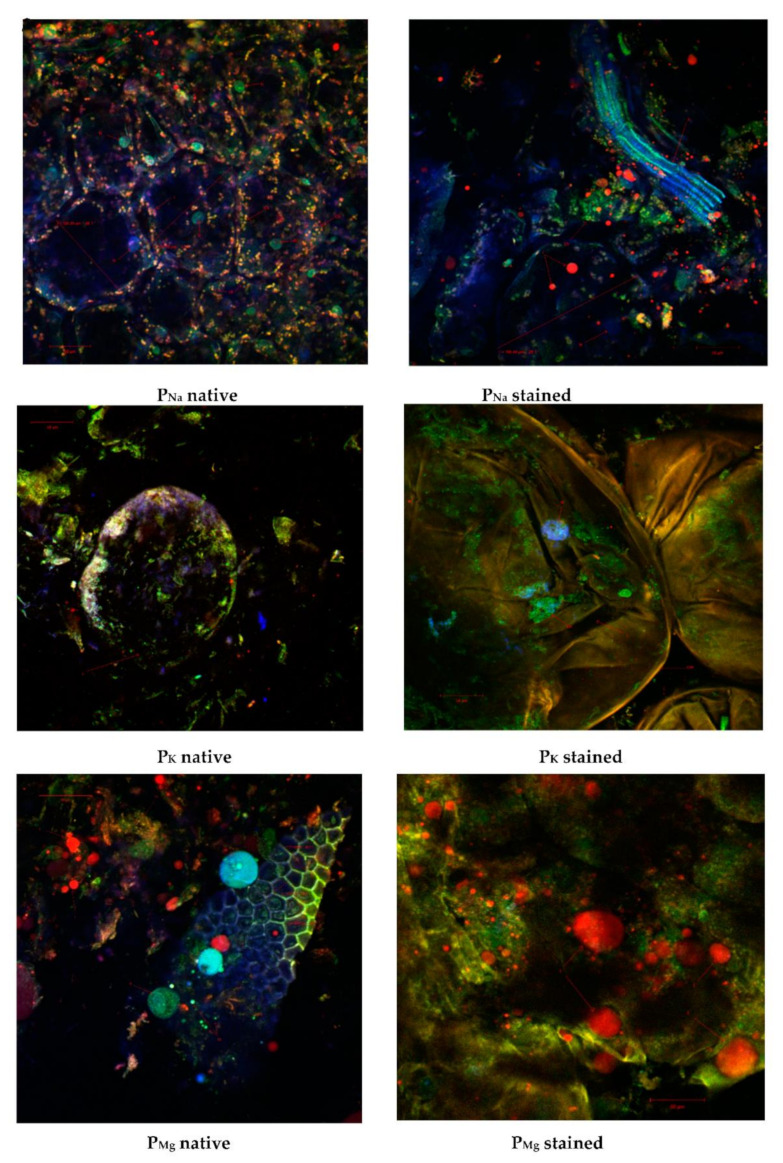
Confocal laser scanning microscopy images of the pickled tomatoes sections obtained with an objective lenses of 20×; scale bar, 50 μm.

**Figure 3 molecules-27-01609-f003:**
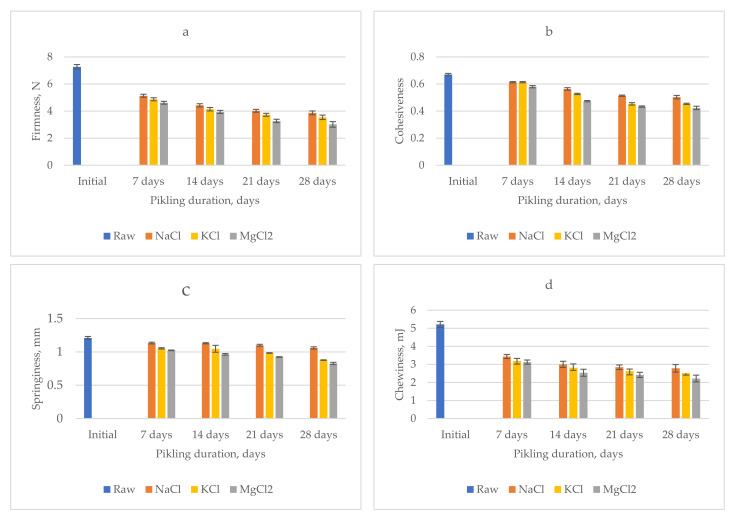
Evolution of textural parameters values during pickling: (**a**) firmness, (**b**) cohesiveness, (**c**) springiness, and (**d**) chewiness.

**Figure 4 molecules-27-01609-f004:**
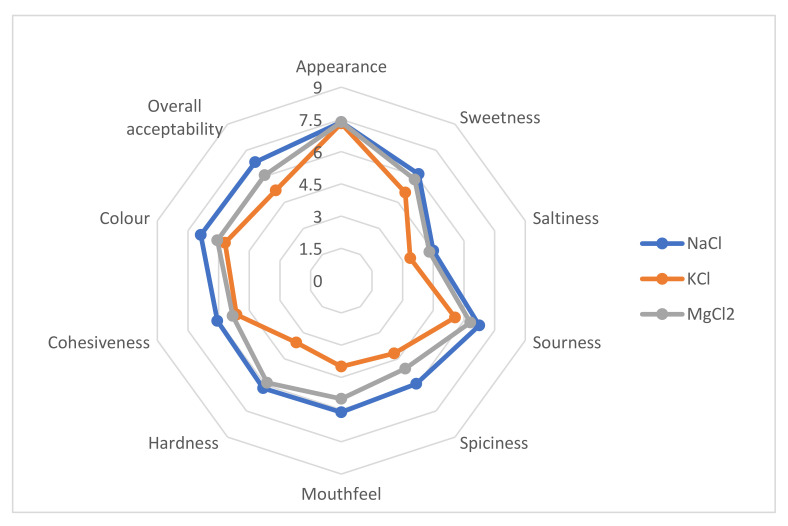
Sensory profile of green pickles tomatoes P_Na_, P_K_, and P_Mg_ fermented for 28 days.

**Table 1 molecules-27-01609-t001:** Physicochemical analysis of the fresh and pickled tomatoes samples.

Sample Code	Moisture, %	Salt, %	Ash, %	Protein, %	Lactic Acid, mg/100 g
**P_0_**	93.65 ± 0.27 ^B^	-	0.37 ± 0.02 ^B^	1.50 ± 0.06 ^B^	-
**P_Na-7_**	88.90 ± 0.16 ^B^	0.55 ± 0.04 ^B^	0.84 ± 0.03 ^B^	1.26 ± 0.07 ^B^	647.38 ± 0.29 ^A^
**P_Na-14_**	90.35 ± 0.15 ^B^	0.73 ± 0.08 ^B^	0.87 ± 0.03 ^B^	1.71 ± 0.04 ^B^	1319.67 ± 0.46 ^A^
**P_Na-21_**	90.35 ± 0.12 ^B^	0.73 ± 0.05 ^B^	0.87 ± 0.04 ^B^	1.56 ± 0.07 ^B^	407.08 ± 0.19 ^A^
**P_Na-28_**	90.88 ± 0.22 ^B^	0.74 ± 0.06 ^B^	0.82 ± 0.05 ^B^	1.53 ± 0.03 ^B^	155.61 ± 0.13 ^A^
**P_K-7_**	91.35 ± 0.12 ^B^	0.70 ± 0.05 ^B^	0.98 ± 0.07 ^B^	1.26 ± 0.07 ^B^	979.58 ± 0.37 ^A^
**P_K-14_**	90.82 ± 0.16 ^B^	0.63 ± 0.04 ^B^	0.83 ± 0.05 ^B^	1.52 ± 0.04 ^B^	1572.77 ± 0.52 ^A^
**P_K-21_**	91.64 ± 0.23 ^B^	0.57 ± 0.06 ^B^	0.69 ± 0.03 ^B^	1.58 ± 0.05 ^B^	565.22 ± 0.18 ^A^
**P_K-28_**	91.47 ± 0.19 ^B^	0.59 ± 0.07 ^B^	0.61 ± 0.04 ^B^	1.79 ± 0.02 ^B^	219.28 ± 0.22 ^A^
**P_Mg-7_**	91.97 ± 0.16 ^B^	0.66 ± 0.02 ^B^	0.88 ± 0.07 ^B^	1.43 ± 0.07 ^B^	405.21 ± 0.45 ^A^
**P_Mg-14_**	91.15 ± 0.11 ^B^	0.66 ± 0.05 ^B^	0.79 ± 0.09 ^B^	1.56 ± 0.04 ^B^	1265.94 ± 0.58 ^A^
**P_Mg-21_**	91.59 ± 0.14 ^B^	0.65 ± 0.07 ^B^	0.69 ± 0.05 ^B^	1.56 ± 0.08 ^B^	373.33 ± 0.26 ^A^
**P_Mg-28_**	91.46 ± 0.12 ^B^	0.62 ± 0.04 ^B^	0.65 ± 0.04 ^B^	1.66 ± 0.09 ^B^	135.17 ± 0.15 ^A^

Values are represented as means ± SD (*n* = 3). Different superscript letters (A and B) mean a significant difference at (*p* > 0.05) among different samples.

**Table 2 molecules-27-01609-t002:** Phytochemical profile of fresh and pickled tomatoes samples.

Sample Code	*β*-Carotene, mg/g DW	Lycopene, mg/g DW	TPC, mg GAE/g DW	TFC, mg CE/g DW
**P_0_**	51.01 ± 0.02 ^C^	34.21 ± 0.05 ^D^	71.66 ± 1.64 ^A^	63.80 ± 0.55 ^A^
**P_Na-7_**	95.23 ± 0.34 ^B,C^	59.18 ± 0.36 ^C,D^	46.89 ± 0.88 ^A^	9.66 ± 0.92 ^B^
**P_Na-14_**	200.15 ± 0.59 ^B^	105.09 ± 0.34 ^B,C^	59.42 ± 0.71 ^A^	20.23 ± 0.56 ^B^
**P_Na-21_**	211.69 ± 0.57 ^A,B^	135.17 ± 0.55 ^A,B^	96.35 ± 0.54 ^A^	34.48 ± 0.66 ^B^
**P_Na-28_**	284.95 ± 0.47 ^A^	162.82 ± 0.82 ^A^	128.03 ± 0.49 ^A^	34.72 ± 0.43 ^B^
**P_K-7_**	128.45 ± 0.32 ^B,C^	83.22 ± 0.12 ^C,D^	45.41 ± 0.54 ^A^	1.80 ± 0.14 ^B^
**P_K-14_**	164.63 ± 0.25 ^B^	115.58 ± 0.38 ^B,C^	57.65 ± 0.85 ^A^	2.81 ± 0.14 ^B^
**P_K-21_**	171.87 ± 0.98 ^A,B^	132.89 ± 0.74 ^A,B^	72.49 ± 0.91 ^A^	6.41 ± 0.79 ^B^
**P_K-28_**	243.11 ± 0.38 ^A^	172.67 ± 0.23 ^A^	72.57 ± 0.91 ^A^	11.87 ± 0.52 ^B^
**P_Mg-7_**	62.04 ± 0.32 ^B,C^	48.18 ± 0.22 ^C,D^	45.59 ± 0.54 ^A^	1.79 ± 0.17 ^B^
**P_Mg-14_**	137.21 ± 0.98 ^B^	70.39 ± 0.66 ^B,C^	57.88 ± 1.09 ^A^	1.81 ± 0.14 ^B^
**P_Mg-21_**	138.72 ± 0.95 ^A,B^	83.11 ± 0.59 ^A,B^	72.78 ± 0.92 ^A^	3.81 ± 0.14 ^B^
**P_Mg-28_**	220.53 ± 1.19 ^A^	124.76 ± 0.80 ^A^	72.79 ± 0.92 ^A^	6.42 ± 0.79 ^B^

Values are represented as means ± SD (*n* = 3). Different superscript letters (A, B, C, and D) mean a significant difference at (*p* > 0.05) among different samples.

**Table 3 molecules-27-01609-t003:** Microbiota of green pickled tomatoes samples during the fermentation.

Type of Analysis	Mesophilic Aerobic Bacteria (log CFU/g)			Yeasts (log CFU/g)			Lactic Acid Bacteria (log CFU/g)		
Sample/Fermentation Time, Days	P_Na_	P_K_	P_Mg_	P_Na_	P_K_	P_Mg_	P_Na_	P_K_	P_Mg_
0	1.31			<1			<1		
7	1.91	2.25	1.69	<1	<1	<1	1.01	1.11	1.21
14	2.05	2.88	1.95	<1	<1	<1	1.56	1.67	1.72
21	2.34	3.14	2.62	<1	<1	<1	3.23	3.56	3.34
28	3.66	3.97	3.76	<1	<1	<1	4.11	4.63	4.32

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
