# Peer review of "The Effect of Sodium Total Substitution on the Quality Characteristics of Green Pickled Tomatoes (Solanum lycopersicum L.)"

_molecules, 2022, doi:10.3390/molecules27051609_

Round 1
Reviewer 1 Report
This paper should be sent to another MDPI journal "Foods".
Author Response
The authors would like to thank the reviewer for the close reading and for the proper suggestions and comments aimed at improving the paper. The present version of the paper has been revised.
Reviewer 2 Report
This paper reports the effect of sodium total substitution on the quality characteristics of green pickled tomatoes.
The detailed the chemical composition, microbiological and sensorial analysis as well as texture profile were interesting.
The subject matter and data are interesting, and paper should eventually be published.
Written nicely but needs some correction.
Line 103 and 221: 16…18 à 16-18
Line 224: 20 - 22 years à 20 - 22 years old
Line 452: the sensory atributes à the sensory attributes
Author Response
The authors would like to thank the reviewer for the close reading and for the proper suggestions and comments aimed at improving the paper. The present version of the paper has been revised.
All the suggestions were made as you mentioned.

Reviewer 3 Report
The manuscript describes the effect of sodium total substitution on the quality characteristics of green pickled tomatoes. It is well organized, with good scientific content and appropriate bibliographic references.
The authors must take into consideration the following remarks and correct several aspects to be considered for publication in Molecules Journal:
- Physico-chemical or physicochemical? Format.
- Abstract: make some considerations about the phisicochemical parameters highlighting the differences on the results concerning NaCl and the other salts.
- Section 2.3.1. Moisture and protein content. At least mention the analytical technique involved in each assay and how the quantification was made (expressed as...). Is protein content given in fresh matter?
- Section 2.3.2. Salt content. The same comment of the previous section.
- Section 2.3.3. Ash content. Ground a wet sample? Moreover, removing a piece of tomato from the marinade and weighted directly can have a higher possibility of having different amount of water than another piece of tomato... Does this determination in fresh matter make sense...?
- Section 2.3.4. Lactic acid determination. Provide the range of concentrations used for the calibration curve and the results are expressed as…
- Section 2.4.1. Mesophilic aerobic bacteria count. Provide the range of concentration used for the dilutions and expressed as log CFU / g of what?
- Section 2.4.2. Yeasts count. More details for this assay. Provide the range of concentration used for the dilutions and expressed as log CFU / g of what?
- Section 2.4.3. Lactic acid bacteria count. Provide the range of concentration used for the dilutions and expressed as log CFU / g of what?
- Section 2.6. Determination of beta-carotene and lycopene content. Is this formula taking into account the mass of dry weight? Please check.
- Section 2.7. Determination of total polyphenol and total flavonoid content. For both methods, calibration curves are required. Please provide the range of concentrations used.
- Section 2.9. Structural and morphological properties of pickled green tomatoes. Line 3. “…using NaCl (P1), KCl (P2) and MgCl2 (P3)”. Correct the abbreviations.
- Section 3.1. Physico-chemical properties of fresh and pickled green tomatoes. Please provide a discussion for the protein content.
- Section 3.2. Effects of pickling on beta-carotene, lycopene, total polyphenol and total flavonoid content. Please provide some considerations concerning lycopene data.
- 3. ABTS•+-radical cation scavenging activity. Eliminate the symbols (duplicate information). Suggestion: ABTS radical cation scavenging activity.
- Figure 1. Please use the same order: Na, K, Mg. Using different fillings in the columns, help to visualize the data. Please highlight the high value of PNa for the 28 days.
- Figure 3. Please use the same order: Na, K, Mg.
- Some of the titles of the articles are with capital letters (e.g. refences 8, 14, 15, 19, etc). Journal name of reference 6 should be in italic. Format them.
Author Response
The authors would like to thank the reviewer for the close reading and for the proper suggestions and comments aimed at improving the paper. The present version of the paper has been revised.
The answers to your recommendations are highlighted with yellow.
Reviewer 4 Report
Green pickled tomatoes are a traditional fermented product in Romania. This study is focused on the effect of total substitution of NaCl with KCl and MgCl2 on physicochemical and microbiological quality, bioactive compounds, microstructural, textural and sensorial properties of fresh and pickled green tomatoes during 28 days of fermentation.
This is an extensive research, with a lot of experimental work.
Thematically the work is interesting for the researchers and professionals and the proposed manuscript is relevant to the scope of the journal.
I found it appropriate for publication in the Molecules journal, but only after some modifications and clarification from the Authors.
The list of keywords could be improved, by adding (or changing) one or two terms.
The overall organization and structure of the manuscript are appropriate. The paper is well written and the topic is appropriate for the journal.
The aim of the paper is well described and the discussion was well approached, its results and discussion are correlated to the cited literature data.
The literature review is comprehensive and properly done.
The novelty of the work must be more clearly demonstrated.
The significance of the Work: Given the large number of analyzed data, this is an interesting study with a possible significant impact in this area.
Statistical interpretation of the analytical data must be more properly presented. Please state which post hoc test was used for investigation of the significance differences between means (Tables 1 and 2)? Perhaps an ANOVA could show the influence of variables (Fig. 1, 2, 3) (Na, Mg, K concentration, time)?
Other Specific Comments: The work is properly presented in terms of the language. The work presented here is very interesting and well done, it is presented in a compact manner.
The methodology applied in the research is presented in clear manner, so that it is repeatable by other authors.
The results are presented in a logical sequence and the discussion and analysis of the results are properly elaborated.
The main drawback of the paper i s the extent of novelty, or the main novelty in the present work, compared to the works of other researchers? In my opinion, the authors should put additional effort to demonstrate that the present work gives a substantial contribution in the research area.
Author Response
The authors would like to thank the reviewer for the close reading and for the proper suggestions and comments aimed at improving the paper. The present version of the paper has been revised.
Statistical interpretation of the analytical data must be more properly presented. Please state which post hoc test was used for investigation of the significance differences between means (Tables 1 and 2)?
At the 2.12. Data analysis (row 261) it is given the test type – Tukey test.
Perhaps an ANOVA could show the influence of variables (Fig. 1, 2, 3) (Na, Mg, K concentration, time)?
An ANOVA one way with the response data in a separate column for each factor level was used for the statistical analysis.
Reviewer 5 Report
The manuscript "The effect of sodium total substitution on the quality characteristics of green pickled tomatoes (Solanum lycopersicum L.)" has an interesting topic. It fits very well in the Molecules journal.
However, there are some open questions that are very important for understanding this manuscript, so it needs to be finalized. The major shortcoming is the methods used for microbiological analyzes.
Section 2.4.1: The authors did not use an adequate method for the enumeration of mesophilic aerobic bacteria. Please explain why did you use an incubation temperature of 37 °C? And why did you chose an incubation time of 48 hours? Please provide a reference. If there is no reference and the authors designed themselves the method, how did they validate it?
The temperature of 37 °C is the optimal temperature for detection of foodborne pathogens such as E. coli, Salmonella spp., L. monocytogenes, S. aureus etc. (please see standards ISO 16649-2:2001, EN ISO 11290-1:2017, EN ISO 6579-1:2017/A1:2020, EN ISO 6888-1:2021) and is not used for enumeration of mesophilic aerobic bacteria. The authors were required to use standards ISO 4833-1:2013 or ISO 4833-2:2013 to determine the number of mesophilic aerobic bacteria.
Additionally, the reference 20 is inappropriate and describes the method for enumeration of yeasts and molds.
Section 2.4.2: The reference 21 is inappropriate and describes the method for enumeration of aerobic bacteria not yeasts and molds. Please provide an adequate reference.
Section 2.4.3: It is unclear why the authors chose an incubation temperature of 43 °C when it is well known that the optimal temperatures for the lactic acid bacteria associated with fruit and vegetable fermentations such us Lactobacillus plantarum, Lactobacillus brevis, Pediococcus pentosaceus and Leuconostoc mesenteroides are 30-35°C, 30 °C, 35 °C and 20-30 °C, respectively. Please explain why did you not use the incubation temperature of 30 °C (prescribed by the standard ISO 15214:1998) used by other authors (see references F.N. Arroyo-López, J. Bautista-Gallego, J. Domínguez-Manzano, V. Romero-Gil, F. Rodriguez-Gómez, P. García-García, A. Garrido-Fernández, R. Jiménez-Díaz (2012). Formation of lactic acid bacteriaeyeasts communities on the olive surface during Spanish-style Manzanilla fermentations. Food Microbiology 32, 295-301; Teresa Zotta, Annamaria Ricciardi, Rocco G. Ianniello, Livia V. Storti, Nicolas A. Glibota, Eugenio Parente (2018). Aerobic and respirative growth of heterofermentative lactic acid bacteria: A screening study. Food Microbiology 76, 117-127; Vasiliki A. Blana, Athena Grounta, Chrysoula C. Tassou, George-John E. Nychas, Efstathios Z. Panagou (2014). Inoculated fermentation of green olives with potential probiotic Lactobacillus pentosus and Lactobacillus plantarum starter cultures isolated from industrially fermented olives. Food Microbiology 38, 208-2018. and many others) or either 37 °C (see reference: Yusuf Alana, Zeynal Topalcengiz, Metin Dığrak (2018). Biogenic amine and fermentation metabolite production assessments of Lactobacillus plantarum isolates for naturally fermented pickles. LWT - Food Science and Technology 98, 322–328).
Author Response
The authors would like to thank the reviewer for the close reading and for the proper suggestions and comments aimed at improving the paper. The present version of the paper has been revised.
The corrections are highlighted with blue.
Section 2.4.1: The authors did not use an adequate method for the enumeration of mesophilic aerobic bacteria. Please explain why did you use an incubation temperature of 37 °C? And why did you chose an incubation time of 48 hours? Please provide a reference. If there is no reference and the authors designed themselves the method, how did they validate it?
The temperature of 37 °C is the optimal temperature for detection of foodborne pathogens such as E. coli, Salmonella spp., L. monocytogenes, S. aureus etc. (please see standards ISO 16649-2:2001, EN ISO 11290-1:2017, EN ISO 6579-1:2017/A1:2020, EN ISO 6888-1:2021) and is not used for enumeration of mesophilic aerobic bacteria. The authors were required to use standards ISO 4833-1:2013 or ISO 4833-2:2013 to determine the number of mesophilic aerobic bacteria.
The ISO 4833-1: 2013 standard was used for mesophilic aerobic bacteria determination. The parameters' corrections were made in the text, as suggested (temperature 30°C, 72h).
Additionally, the reference 20 is inappropriate and describes the method for enumeration of yeasts and molds.
Section 2.4.2: The reference 21 is inappropriate and describes the method for enumeration of aerobic bacteria not yeasts and molds. Please provide an adequate reference.
The references for these two methods were accidentally reversed in the list of references. The necessary corrections were made; thus, the standard ISO 4833-1: 2013 (reference 20) was used for the mesophilic aerobic bacteria and the standard ISO 21527-2: 2008 (reference 21) for the yeasts and molds.
Section 2.4.3: It is unclear why the authors chose an incubation temperature of 43 °C when it is well known that the optimal temperatures for the lactic acid bacteria associated with fruit and vegetable fermentations such us Lactobacillus plantarum, Lactobacillus brevis, Pediococcus pentosaceus and Leuconostoc mesenteroides are 30-35°C, 30 °C, 35 °C and 20-30 °C, respectively. Please explain why did you not use the incubation temperature of 30 °C (prescribed by the standard ISO 15214:1998) used by other authors (see references F.N. Arroyo-López, J. Bautista-Gallego, J. Domínguez-Manzano, V. Romero-Gil, F. Rodriguez-Gómez, P. García-García, A. Garrido-Fernández, R. Jiménez-Díaz (2012). Formation of lactic acid bacteriaeyeasts communities on the olive surface during Spanish-style Manzanilla fermentations. Food Microbiology 32, 295-301; Teresa Zotta, Annamaria Ricciardi, Rocco G. Ianniello, Livia V. Storti, Nicolas A. Glibota, Eugenio Parente (2018). Aerobic and respirative growth of heterofermentative lactic acid bacteria: A screening study. Food Microbiology 76, 117-127; Vasiliki A. Blana, Athena Grounta, Chrysoula C. Tassou, George-John E. Nychas, Efstathios Z. Panagou (2014). Inoculated fermentation of green olives with potential probiotic Lactobacillus pentosus and Lactobacillus plantarum starter cultures isolated from industrially fermented olives. Food Microbiology 38, 208-2018. and many others) or either 37 °C (see reference: Yusuf Alana, Zeynal Topalcengiz, Metin Dığrak (2018). Biogenic amine and fermentation metabolite production assessments of Lactobacillus plantarum isolates for naturally fermented pickles. LWT - Food Science and Technology 98, 322–328).
The observation is correct; we used an incubation temperature of 35±2°C. The error in the text was corrected.
Round 2
Reviewer 1 Report
The manuscript has been corrected according to the Reviewers comments. I find it suitable for publication in the Molecules journal.
Reviewer 5 Report
The authors have provided a nicely detailed and thorough response to the comments from the previous review and have addressed my major concerns. If the microbiological examinations really performed with the appropriate methods, and if only a technical error occurred during manuscript writing and referencing methods, I believe that the work should be accepted in its current form.